# A Preliminary Study of the Influence of High Intensity Laser Therapy (HILT) on Skin Surface Temperature and Longissimus Dorsi Muscle Tone Changes in Thoroughbred Racehorses with Back Pain

**DOI:** 10.3390/ani13050794

**Published:** 2023-02-22

**Authors:** Paulina Zielińska, Maria Soroko-Dubrovina, Krzysztof Dudek, Iliana Stefanova Ruzhanova-Gospodinova

**Affiliations:** 1Department of Surgery, Faculty of Veterinary Medicine, Wroclaw University of Environmental and Life Sciences, Plac Grunwaldzki 51, 50-366 Wrocław, Poland; 2Institute of Animal Breeding, Wroclaw University of Environmental and Life Sciences, Chelmonskiego 38C, 51-630 Wrocław, Poland; 3Center for Statistical Analysis, Wroclaw Medical University, Marcinkowskiego 2-6, 50-368 Wrocław, Poland; 4Department of Anatomy, Physiology and Animal Sciences, University of Forestry, 1797 Sofia, Bulgaria

**Keywords:** high-intensity laser therapy, Kissing Spines Syndrome, back pain, thermography, thoroughbreds, racehorse

## Abstract

**Simple Summary:**

Back pain in Thoroughbred racehorses is frequent and significantly decreases their athletic performance. The most common thoracolumbar alteration in Thoroughbreds is Kissing Spines Syndrome (KSS). The objective of the current study was to evaluate and compare soft tissue response to high-intensity laser therapy (HILT) by measuring changes in skin surface temperature and longissimus dorsi muscle tone, located in the thoracolumbar back area, in Thoroughbreds with back pain diagnosed with and without KSS. The Thoroughbreds were divided into two groups, those with KSS (n = 10) and those without KSS (n = 10). A single laser treatment of the longissimus dorsi muscle (on the left side, between the fifteenth thoracic and the second lumbar vertebrae) was performed. Thermographic examination and palpation were repeated before and after HILT to assess changes in skin surface temperature, muscle tone and pain response. In both groups, HILT was associated with an average skin surface temperature increase of 2.5 °C and a palpation score reduction of 1.5 points, without any differences between the groups. In conclusion, HILT was found to be a safe and supportive treatment method for longissimus dorsi muscle pain and discomfort as assessed by digital palpation in Thoroughbreds. The results of the present study are encouraging, but further studies with larger samples, a longer follow-up period and comparisons with placebo control groups are needed to draw a more valid conclusion.

**Abstract:**

The reason for undertaking this study was to investigate soft tissue response to high-intensity laser therapy (HILT) by measuring changes in skin surface temperature and longissimus dorsi muscle tone in the thoracolumbar back area in Thoroughbreds with back pain and diagnosed with and without Kissing Spines Syndrome (KSS). Thoroughbreds aged 3–4 years with clinically presented back pain underwent a radiological examination (to assess a lack or presence of KSS) and longissimus dorsi muscle palpation (to assess muscle tone and pain degree). The subjects were divided into two groups, those with KSS (n = 10) and those without KSS (n = 10). A single HILT treatment on the longissimus dorsi muscle, on the left side, was performed. Thermographic examination and palpation were repeated before and after HILT to assess changes in skin surface temperature and muscle pain response. In both groups, HILT caused a significant increase in skin surface temperature of 2.5 °C on average and a palpation score reduction of 1.5 degrees on average (*p* = 0.005 for both measurements), without differences in any outcome measures between the groups. Furthermore, the correlation between changes in the average skin surface temperature and the average palpation scores in horses with and without KSS were negative (rho = 0.071 and r = −0.180, respectively; *p* > 0.05). The results of the present study are encouraging, but further studies with larger samples, a longer follow-up period and comparisons with placebo control groups are needed to draw a more valid conclusion.

## 1. Introduction

Thoracolumbar back pain is an important factor causing poor performance in ridden horses [1,2]. A horse’s back protects and stabilizes the body frame and is composed of both soft tissue, such as the muscles and ligaments, and the rigid skeletal elements, which create functional consistency. Equine back disorder may be the result of their individual conformation. Soft tissue injuries are often found in long-backed horses, while osseus lesions are diagnosed in short-backed animals [3]. The longissimus dorsi muscle and supraspinous ligament are commonly found to be the soft tissue cause of thoracolumbar pain [3]. The main cause of epaxial muscle and ligament pain is improper saddle and girth fitting [4], the animal’s emotional tension or physiological stress [5], and the rider’s load on the horse’s back [6,7]. One of the most common osseus causes of back pain is dorsal spinous process impingement, which is a part of overriding dorsal spinous processes, often referred to as “Kissing Spines Syndrome” (KSS) [8]. KSS can be diagnosed in all breeds, at any age, and any sex [1], but racing Thoroughbreds have been shown to have a higher prevalence when compared with other breeds [9]. KSS is mostly localised between the T10-T18 vertebrae, but it can also affect the lumbar spine [10]. One of the major clinical sings of KSS is chronic longissimus dorsi muscle soreness and increased muscle tone. The pathway of secondary epaxial muscle pain, associated with primary osseus lesions in horses, is not understood in detail. One possible explanation, based on human medicinal knowledge, is that spinal vertebrae diseases cause local ipsilateral multifidus muscle atrophy, resulting in spinal instability. The musculoskeletal system then compensates for the disability by epaxial muscle tension and shortening [11,12].

The clinical signs of back pain are highly variable [9], but orthopedic evaluation of muscle pain in horses is traditionally assessed by palpation [13]. Deep palpation allows for a general evaluation of back health, based on spinal mobilization and palpable muscle hypertonicity trigger points [14,15]. Pain is detected based on the evaluation of “pain reactions”, which are measured in terms of several scoring systems and scales used by both scientists and practitioners [13,16,17].

Longissimus dorsi pain, whether of primary or secondary origin, reduces thoracolumbar spine flexibility and, thus, disrupts its proper biomechanics [10]. The elimination of epaxial muscle spasms and pain is therefore an important part of treating back diseases in horses.

Several treatment modalities are available for treating epaxial muscle pain in horses. Medical therapies include the use of general administered non-steroidal anti-inflammatory drugs [3,8,10] or local corticosteroid injections [8,10]. Alternative treatments include manual therapy such as massage, stretching or chiropractic, and physical therapies like magnetic field therapy, extracorporeal shock wave therapy and hydrotherapy [18,19,20]. Currently, laser therapies such as high-intensity laser therapy (HILT) are very popular [21]. HILT is a non-invasive and safe therapy based on the application of focused light generated by class IV (power > 0.5 W) laser devices [22]. It utilises high peak power (1–3 kW), infrared wavelengths (600–1000 nm) and short single pulse durations (>150 ms). As such, HILT can reach deep tissue without excessive thermal effects or cellular damage and can be used in the treatment of bigger joints and large muscle groups, which are difficult to reach with low power lasers [22,23]. Only a few equine studies have demonstrated the positive results of HILT in the treatment of musculoskeletal disorders. HILT is effective in the treatment of soft tissue injuries, such as tendinopathies and desmopathies [24,25,26,27,28]. In an earlier article, we described eleven clinical cases of horses with tarsal osteoarthritis, treated with HILT as a monotherapy, where we found that HILT reduced joint pain and lameness grade, but poorly limited joint discomfort after a flexion test in short-term outcomes [29]. 

Studies describing the impact of laser therapy on muscle pain and tension have been conducted in both experimental animals and humans. Lopes-Martins et al. [30] have found that 0.5 J/cm^2^ and 1 J/cm^2^ (655 nm and 2.5 mW) doses of low-level laser therapy prevent the development of muscle fatigue in rats after repeated tetanic contractions. Ramos et al. [31] have reported that 3 J doses of low-level laser therapy at 810 nm and 100 mW are effective for improving functional outcomes in the early phase following tibialis anterior muscle strain injury in rats. In humans, studies have shown that the photothermal effects of HILT may lead to improved muscle relaxation which, thus, reduces pain [32]. Moreover, HILT (808 nm wavelength, continuous wave mode, average power of 4.28 W/cm^2^) has resulted in positive effects in relieving muscle tension in patients with hemiplegia [33]. The only research which evaluated the clinical effectiveness of laser therapy in treating thoracolumbar pain in horses was performed by Haussler et al. [34]. They found that laser treatment produced significant reductions in back pain, epaxial muscle hypertonocity and trunk stiffness in 61 competitive western performance horses. However, this research involved the use of low-level laser therapy. 

A review of the published literature was carried out using the following databases: Scopus, Web of Science and Google Scholar. Additionally manual searches were performed in the indices of the *Equine Veterinary Journal* and *Journal Equine Veterinary Science*. The key words for the literature search included “HILT”, “High Intensity Laser Therapy”, “musculoskeletal pain”, “muscle pain”, “Kissing Spines Syndrome”, and “Thoroughbreds”. To the best of our knowledge, this study is the first clinical examination of the effectiveness of HILT in treating muscle tissue in horses with back muscle pain. The objectives of the study were to evaluate and compare soft tissue responses to HILT by measuring changes in skin surface temperature and longissimus dorsi muscle tone in the thoracolumbar back area in Thoroughbreds presenting back pain (diagnosed with or without KSS). It was hypothesised that HILT would increases skin surface temperature and reduce longissimus dorsi muscle tone in both groups of horses but that better effects would be achieved in horses without KSS.

## 2. Materials and Methods

The Animal Welfare Advisory Team at Wroclaw University of Environmental and Life Sciences approved the study design, which is in compliance with Polish and European Union legislation on animal experimentation (no 1/2023). The procedures used in this study were deemed not to cause pain, suffering, distress or lasting harm equivalent to or higher than that caused by the introduction of a needle (article 1.5f EU directive 2010/63/EU). Ethical approval was granted without a formal application. Written consent was obtained from Partynice Racecourse in Wroclaw for all the racehorses participating in this study.

### 2.1. Horses and Inclusion Criteria

The horses were selected from the Partynice Racecourse in Wroclaw (Poland) in September 2021. A total of 20 3–4 year-old racehorses (11 stallions and 9 mares) were selected after fulfilling the inclusion criteria. The criteria were as follows: (1) present longissimus dorsi muscle pain in thoracolumbar region palpation; (2) be healthy according to basic clinical examination and to have no clinical lameness in walking and trotting on hard ground in a straight line; (3) maintain the same race training programme with the same trainer; (4) be free from any systemic and local administration of anti-inflammatory drugs or analgetic drugs during the 8 weeks before the study; (5) be free from any manual or physical therapy, including acupuncture, massage, osteopathy or magnetotherapy in the 4 weeks prior to the study; (6) have the presence or lack of radiographic signs of KSS (radiological assessment and KSS criteria are described below); (7) have the presence of pigmented skin in the thoracolumbar back area (black, bay or chestnut coat color).

### 2.2. Study Design

The horses were divided into two groups, those with KSS (n = 10) and those without KSS (n = 10). On the examination day, each horse underwent the same examination protocol. The horses were inspected early in the morning and before any activity. All the measurements were conducted with the animal standing equally on all weight bearing limbs in a stable corridor. Thermographic examination was performed to determine the skin surface temperature of the thoracolumbar back, followed by longissimus dorsi muscle palpation, to assess tone and pain degree. A single laser treatment of the longissimus dorsi muscle was then performed. The thermographic examination followed by palpation were then repeated immediately after HILT to assess changes in skin surface temperature and muscle pain response. 

### 2.3. Radiographic Examination and Palpation

Laterolateral projections of the thoracolumbar spine were performed for the purposes of radiological KSS assessment. The horses stood square and weightbearing on hard ground, with the head and neck in a natural position to avoid false changes in the distance between the dorsal spinous processes [35]. The images were blindly and retrospectively evaluated by one veterinarian (P.Z.). According to Turner [9], a diagnosis of KSS can be made when two or more vertebrae touch or overlap. 

Non-instrumental palpation, i.e., conventional palpation, was performed unilaterally (left side) on the longissimus dorsi muscle in the area between the fifteenth thoracic and the second lumbar vertebrae. A palpation scoring system for horse muscle tone and pain reaction was used, according to Varcoe-Cocks et al. [13], which is as follows: (0) soft, with low muscle tone; (1) normal tone; (2) stiff muscle but not painful; (3) stiff and/or painful muscle with slight associated spasms but without horse movement; (4) painful muscle with associated spasms and local horse movement, i.e., pelvic tilt and extension response; (5) very painful muscle with spasm and behavioural response, i.e., ears flat back and kicking. The horses were examined and scored subjectively for longissimus dorsi muscle pain by a qualified equine physiotherapist (M.S.D.) 

### 2.4. Thermographic Examination 

The thermographic examination was performed with a VarioCam HR infrared camera (uncooled microbolometer focal plane array; resolution, 640 × 480 pixels; spectral range, 7.5–14 mm; accuracy ±1 °C, sensitivity 0.02 °C InfraTec, Dresden, Germany). Prior to the study, the thermal camera was quality assured using an Isotech 988 blackbody calibration source (Isothermal Technology, Stockport, United Kingdom). The examination protocol was the same as previously described by Soroko et al. [36]. To reduce the effect of environmental factors, like air draughts and sunlight, the images were obtained within an enclosed stable with closed windows. Horses were examined with the acclimatization period of 30 min prior the imaging and had a brushed back area, approximately 1 h before examination. The distance between the horse’s back and the equipment was set at 1.5 m for all images, with the emissivity set at 1 for all measurements [37]. The average ambient temperature in the stable was 19 °C and humidity 45% at the time that the images were taken, as measured by a TES 1314 thermometer (TES, Taipei, Taiwan). Regions of interest (ROIs) were determined for each thermographic image. The average skin surface temperature of the square treatment area (10 × 10 cm^2^) in the region under investigation was determined using IRBIS 3 Professional software (InfraTec, Dresden, Germany) (Figure 1). The thermographic examination was performed by the same therapist (M.S.D.).

### 2.5. High Intensity Laser Therapy

The laser therapy was performed with a class IV Polaris HPS laser (Astar, Bielsko-Biała, Poland), a commercial laser light source used in human physiotherapy and in veterinary medicine. The Polaris HPS has two synchronised sources of different wavelength laser light in the near-infrared spectrum. The first wavelength is 808 nm (AlGaAs laser with 8 W of output power), while the second is 980 nm (InGaAs/AlGaAs laser with 10 W of output power). The two wavelengths are emitted simultaneously with the propagation axes of the two laser beams being coincident. The same parameters were used at both wavelengths. The energy density was 20 J/cm^2^, power was 5 W, frequency was 100 Hz and duty cycle was 80%. The total energy delivered over a period of 500 s was 2000 J. The square treatment area was the same as the region previously identified as an ROI in the thermographic examination and was localised over the longissimus dorsi muscle in the thoracolumbar junction on the horse’s left side (in the area between the fifteenth thoracic and the second lumbar vertebrae). The treatment area was not shaved, and no other skin preparation was performed. The laser treatment was administered using a handpiece held in firm contact with the tissue and manually moved during treatment while maintaining even irradiation of the treatment surface. The handpiece spot size was 5 cm^2^. Both the person holding the horse and the therapist wore laser goggles. The laser scanning in all horses was performed by the same veterinarian (P.Z.).

### 2.6. Statistical Analysis

Statistical analysis of the results obtained was performed using STATISTICA v. 13.3 (TIBCO Software Inc., Palo Alto, CA, USA) and a MS EXCEL template (Microsoft, Redmond, Washington, USA). The data were first plotted for normal distribution analysis using the Shapiro–Wilk test. The significance level was set at *p* < 0.05. For the quantitative continuous and discrete variables, the following basic descriptive statistics were estimated: medians (Me), lower (Q1) and upper (Q3) quartiles, extreme lowest values (Min) and extreme highest values (Max). Depending on the test results, the non-parametric Wilcoxon signed-rank test was used to compare baseline and post-treatment within a group, while Mann–Withney’s *U* test was used for comparison between the groups. Correlations between differences in skin surface temperatures and palpation score after the application of HILT in the two groups were calculated using Spearman’s rank correlation coefficient (rho).

## 3. Results

The average skin surface temperatures and average palpation scores in horses diagnosed with and without KSS, both before and after HILT, did not differ significantly (*p* > 0.05; Table 1). 

In both groups, HILT treatment caused average skin surface temperature to increase by 2.5 °C (Z = 2.803, *p* = 0.005, Table 1, Figure 2) and a palpation score reduction of 1.5, which gave a highly significant difference (Z = 2.803, *p* = 0.005, Table 1, Figure 3).

There was no correlation between changes in the average skin surface temperature and average palpation scores in either group (rho = 0.071 for group without KSS and r = −0.180 for with KSS, *p* > 0.05; Table 2).

## 4. Discussion

Horses are predisposed to general back pain and back disorders because of the type and intensity of their work [38]. Furthermore, a study carried out on 572 horses showed that Thoroughbreds have a 76% greater prevalence as well as a higher KSS grading system than other breeds [39]. This is probably because the tips of their spinous processes are close to each other [40]. The elimination of epaxial muscle spasms and pain (whether it is primary or secondary) plays an important role in the welfare of racehorses.

In this small trial, the hypothesis that HILT increases skin surface temperature and reduces longissimus dorsi muscle sensitivity in horses with and without KSS was confirmed, but the groups did not differ in terms of the parameters measured, as we expected. Moreover, there was no correlation between changes in the average skin surface temperature and the average palpation score in either group.

Similar findings, regarding skin surface temperature increases after HILT, have been reported in our previous study performed on clinically healthy tissue. A single HILT irradiation of the dorsomedial aspect of the tarsal joint in 16 racing Thoroughbreds caused a significant increase in skin surface temperature in the treatment area [41]. In both this study and the previous one, the mean temperature increase was 2.5 °C. The laser parameters used in our previous study were lower compared to those used in the current study. A possible explanation for the similar thermal effect in both studies might be found in regard to the preparation of the irradiated area. In our previous research, the treatment area was additionally clipped. It is known that when laser light passes through a coat it can cause reflection, absorption and a scattering of photons, thus reducing photon penetration in the target tissue [42,43,44]. The photothermal effect of HILT induces mild local hyperthermia sufficient enough to accelerate cellular activity and vasodilatation, thus leading to enhanced local blood circulation in irradiated tissue [45]. This results in a hastened removal of inflammatory cytokines, improvement in the mitochondrial oxidation process, production of adenosine triphosphate, and more efficient absorption of tissue swelling [46]. In the current study, we did not control the presence and degree of vasodilatation, although our previous research confirmed vessel diameter increase in irradiated tissue immediately after HILT [47,48].

Mild inflammation is reduced when tissue temperature increases by more than 1 °C, and an analgesic effect and muscle relaxation is obtained when it increases by 2–3 °C. Changes in tissue extensibility can be reached with a temperature increase of 3–4 °C [49]. Muscle sensitivity is one of the main causative factors in decreased muscle flexibility in horses diagnosed without KSS. Conversely, both bone tissue and inflamed muscles with the connective tissue are the main targets in the treatment of horses diagnosed with KSS. The present study showed a palpation score reduction in the longissimus dorsi muscle of 1 to 2 degrees in both groups of horses. In their prospective study, Alayat et al. [32] also confirmed that the photothermal effect leads to improved fascia extensibility and a muscle-relaxing effect and, thus, a reduction in muscle sensitivity. Similar results have been found in human hemiplegic patients, where HILT rapidly decreased muscle tension during laser irradiation, with low muscle tension maintained following treatment [34]. 

Muscle relaxation and muscle sensitivity reduction are also the outcome of the photomechanical effect of HILT [22]. A particular waveform with regular peaks of elevated values of amplitude and distances (in time) between them can rapidly induce in the deep tissue a photomechanical effect [50]. HILT can provide extremely brief beats at a maximum repetition rate, thus creating real pressure. Heat waves travel through the tissue, stimulating the free nerve endings and causing their inhibition [51]. 

A possible explanation for the lack of differences between the groups in longissimus dorsi muscle sensitivity reduction is the short-term HILT efficacy control. It can be assumed that, in horses without KSS, the myorelaxation effect can be longer than in horses with KSS, which (as a chronic disease) causes constant tissue irritation. Lee et al. [33] have pointed out that in humans with hemiplegia, treatment efficacy decreases exponentially as a function of illness duration. Moreover, many other studies have shown that the effectiveness of treatment decreases with the duration of the disease [30,52]. In our study, we did not clarify the underlying cause of pain in horses without KSS. Furthermore, in the horses with KSS, we have not established the real cause of back pain. Further studies are needed to clarify this matter. 

None of the horses included in the study experienced skin burns, swelling or pain reactions during or after HILT application. Moreover, all horses had no interruption in their daily race training due to complications after HILT. The maximum temperature reached after HILT was 34.5 °C, while superficial thermal injuries are noted after topical application of heat at 50 °C [53]. The use of different HILT parameters, with different protocols, may be necessary in future research investigating the maximum positive effects of laser therapy, without tissue overheating.

The main limitation of this study was the lack of long-term assessment, especially in horses diagnosed with KSS. The other limitation included a lack of control or laser-sham group, and a blinding test was not included for the post-treatment data collection. There is need for follow-up data to investigate the number and frequency of treatments needed to achieve the best therapeutic effects. The relatively small sample size in both groups of horses was also a study limitation. It would be ideal to have a larger number of Thoroughbreds presenting with back pain, regardless of whether they have KSS. However, from a clinical point of view, it was possible to indicate the positive impact of HILT in the case of treatment performed on a large muscle, such as the longissimus dorsi muscle. Automatic laser scanning is also recommended in future research to overcome the limitations of manual scanning, although the same laser operator was used in all applications.

## 5. Conclusions

In conclusion, HILT is a safe and supportive treatment method for longissimus dorsi muscle pain and discomfort in Thoroughbreds under the conditions of this study. Furthermore, the photothermal and muscle-relaxing effects were similar in horses suffering from longissimus dorsi muscle pain, regardless of the presence of KSS. The results of the present study are encouraging, but further blinded studies with larger samples, longer follow-up periods and possible comparisons with placebo control groups are needed to make a more valid conclusion.

## Figures and Tables

**Figure 1 animals-13-00794-f001:**
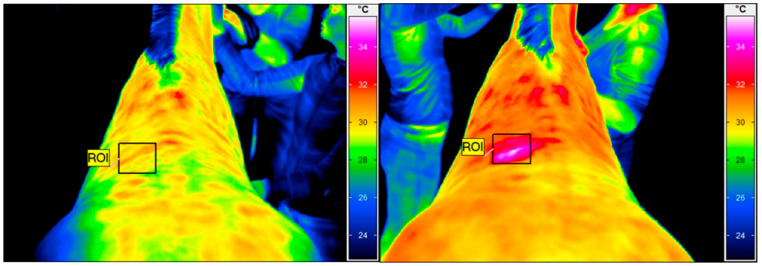
Thermographic images of the thoracolumbar back region. The square area (ROI) indicates the skin surface temperature before HILT (**left**), with an average temperature of 29.8 °C, and immediately after (**right**), with an average temperature of 32.9 °C.

**Figure 2 animals-13-00794-f002:**
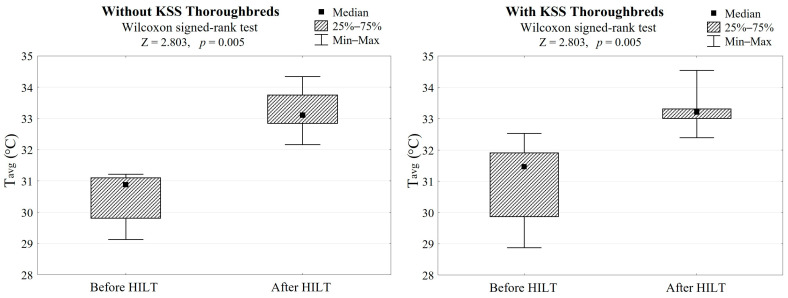
Differences in skin surface temperatures after HILT application in both groups (with/without KSS) and the Wilcoxon signed-rank test results. *T*_avg_ refers to the average skin surface temperature after HILT.

**Figure 3 animals-13-00794-f003:**
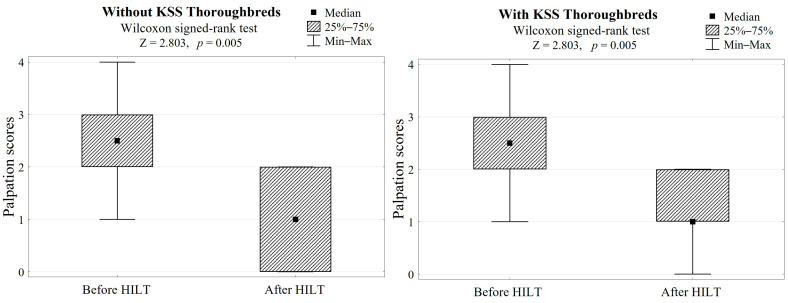
Differences in palpation scores after HILT application in both groups (with/without KSS) and the Wilcoxon signed-rank test results.

**Table 1 animals-13-00794-t001:** Skin surface temperature, palpation score and differences in skin surface temperature and palpation score (median and IQR) before and after HILT treatment for horses diagnosed with and without KSS.

Parameters	Without KSS (n = 10)	With KSS (n = 10)	Without KSS vs. With KSS (*p*-Value)	Power 1 - β
*T*_avg_ before HILT (°C)	30.9 (29.8 to 31.1)	31.5 (29.9 to 31.9)	0.140	0.974
Min–Max	29.1–31.2	28.9–32.5		
*PS* before HILT (score)	2.5 (2 to 3)	2.5 (2 to 3)	0.969	0.051
Min–Max	1–4	1–4		
*T*_avg_ after HILT (°C)	33.1 (32.8 to 33.8)	33.2 (33.0 to 33.3)	0.734	0.562
Min–Max	32.2–34.3	32.4–34.5		
*PS* after HILT (score)	1 (0 to 2)	1 (1 to 2)	0.601	0.796
Min–Max	0–2	0–2		
Δ*T*_avg_ (°C)	2.6 (2.3 to 3.1)	2.4 (1.3 to 3.3)	0.307	0.863
Min–Max	1.4–3.8	0.7–3.5		
Δ*PS* (score)	−2 (−2 to −1)	−1 (−2 to −1)	0.322	0.999
Min–Max	−2–−1	−3–−1		

Abbreviations: HILT, high intensity laser therapy; KSS, kissing spines syndrome, *T*_avg_, average skin surface temperature; *PS*, palpation scores; Δ*T*_avg_, differences in values of average skin surface temperature after HILT; Δ*PS*, differences in values of palpation scores after HILT.

**Table 2 animals-13-00794-t002:** Correlation between changes in average skin surface temperatures (Δ*T*_avg_) and changes in palpation scores (Δ*PS*) after application of high-intensity laser therapy (HILT) in groups with Kissing Spines Syndrome (KSS) and without KSS and results of Spearman’s rank correlation coefficient (rho).

Statistics	Without KSS (n = 10)	With KSS (n = 10)	Without KSS and With KSS (n = 20)
Rho (95% CI)	0.071 (−0.585 to 0.671)	−0.180 (−0.727 to 0.507)	−0.064 (−0.493 to 0.389)
*p*-value	0.831	0.590	0.780

## Data Availability

Data is contained within that article.

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
