# Peer review of "A Preliminary Study of the Influence of High Intensity Laser Therapy (HILT) on Skin Surface Temperature and Longissimus Dorsi Muscle Tone Changes in Thoroughbred Racehorses with Back Pain"

_animals, 2023, doi:10.3390/ani13050794_

Round 1
Reviewer 1 Report
This well-written paper addresses local response to the application of high intensity laser light in thoroughbred horses with sore backs. The thoroughness of the literature review is noted and appreciated. The study presented here attempts to differentiate between response in horses with and without kissing spine syndrome. By experimental design it addresses only short-term/acute responses. It adds little to previous work performed by the authors and others, while its design leaves unanswered questions. Its conclusions are only partially supported by its findings, some of which are suspect. My concerns are detailed below.
No controls were employed and no sham treatments were used. The team member who performed digital palpation knew that the horses had been treated immediately before repeat palpation - they were not blinded to treatments. These two problems essentially invalidate the palpation data, even though there may well have been some response in terms of sensitivity to palpation. The data most certainly do not have value in objectively predicting potential benefits of the treatment.
The paper frequently engages in extrapolation to clinical application and treatment efficacy, though no evaluation of clinical efficacy was performed beyond response to digital palpation shortly after treatment. Potential clinical benefits are posited largely through reference to experimental studies in other species, and are theoretical rather than demonstrated. The observations made by the researchers may parallel their clinical experience in the field use of the technique, but clinical evaluation beyond the measurements made were not part of the experimental design in this study and claims of efficacy cannot be made.
More information is needed on environmental controls employed during collection of thermographic data - were all data collected under controlled conditions and out of sunlight, for example? Were drafts excluded, sources of radiant heat, what was the time of year? Were thermographic devices adequately calibrated? What was the difference between the stall and corridor environments, what time was allowed for stabilisation?
The authors describe a lack of long-term assessment as the main limitation. I would add failure to include controls and to blind assessors to treatments as equally large if not more significant shortcomings. Long-term assessment would be important if the therapeutic effect was being assessed, which is not the case in this acute study of local effects.
Conclusions - the first conclusion is not justified based on the information presented. The study design does not allow these questions to be answered. The last conclusion is accurate. No blinding was employed - the evaluator performing palpation was aware of the treatment status of the horses and no controls were employed. There is thus a huge potential for bias, one that cannot be assessed from the data presented.
The terms muscle tone, muscle pain (a.k.a. impact on locomotion) and sensitivity (response to digital palpation) tend to be used synonymously in this paper. This is both confusing and questionable. There was no assessment of the horses' movement, numbers are small, only one operator performed palpation on one occasion (before and after treatment) and other circumstances may not have been fully controlled, despite the inclusion criteria, in a working stable. Despite this the authors state that the treatment is effective and efficient, yet have no firm basis for drawing this conclusion and did not in fact attempt to measure a sustained treatment effect. It might be better if they confined themselves strictly to the essential elements of the experimental protocol - skin temperature and palpation scores, together with perhaps absence of difference between the two types of case. Comments referencing the potential implications of their observations to routine use of the procedure can be made in the discussion, with the limitations being more fully emphasised there.
Line 20 - I suggest the word "caused" might be better changed to "was associated with". Perhaps the whole Simple Summary might be simplified for consumption by a lay audience.
Line 21 - " degrees" should be replaced with "points", lest the palpation score be confused with temperature changes.
Line 23 - suggest the phrase "as assessed by digital palpation," be inserted after "discomfort"
Line 22 - this study did not perform any assessments that would justify use of the term "efficient".
Line 40 - this statement is not substantiated
Line 60 - hypertension - I assume you are referencing increased tension in the muscle? I suggest changing the word to avoid confusion with vascular hypertension.
Line 152 - change "Tuner" to "Turner"
Line 213 - why are differences between groups at baseline described as "no statistically significant differences" yet comparison between the two study groups is referred to by the term "non-significant differences"? Statements imply a difference between the two comparisons - there is none, neither were statistically significant.
Line 254 - as noted, not substantiated by the findings
Line 283 - the sentence must be removed, this study did not achieve this outcome
Line 293 - for how long were the horses monitored afer treatment, surely the statement cannot be made unless the protocol included follow-up for at least 48 hours?
Reviewer 2 Report
This study reports the effects of high intensity laser therapy on the clinical signs of horses presented for investigation of back pain.
High-intensity laser therapy has been shown to be of value in the treatment of various orthopaedic diseases in horses. Investigation of its applicability to the treatment of back pain is therefore justified.
Introduction.
The introduction provides a good overview of the problem of back pain in horses and the justification for the use of high-intensity laser therapy in its treatment.
The authors comment that “HILT can reach deep tissue without excessive thermal effects or cellular damage and can be used in the treatment of bigger joints and large muscle groups, which are difficult to reach with low power lasers.” It would be worth developing this statement based on the findings of Haussler et al. He reported that low intensity laser therapy had a beneficial effect on back pain in horses.
Kevin K. Haussler, Philippe T. Manchon, Josh R. Donnell, David D. Frisbie. Effects of Low-Level Laser Therapy and Chiropractic Care on Back Pain in Quarter Horses. Journal of Equine Veterinary Science. Volume 86, 2020, 102891, https://doi.org/10.1016/j.jevs.2019.102891.
Specific comments on the introduction
“Soft tissue injuries are often found in long-backed horses, while osseus lesions are diagnosed in short-backed animals [3].”
Presumably this comment refers to the confirmation of individual horses. It would be helpful if the meaning of the sentence could be clarified.
“One of the major clinical sings of KSS is chronic longissimus dorsi muscle soreness and hypertension.”
The term hypertension is generally related to elevated blood pressure. Could another word be found to better describe what the authors are reporting?
Methods.
The methods are appropriate, using previously published protocols in most cases. Further detail about the scoring of back pain by the physiotherapist would be useful.
“The square treatment area was the same as in the thermographic examination and was localised over the longissimus dorsi muscle, in the thoracolumbar junction on the horse’s left side (in the area between the fifteenth thoracic and the second lumbar vertebrae).” This is an important point so it should be made clear that the therapeutic light was applied to the region previously identified as an ROI on thermographic examination.
Was blinding used for the collection of post-treatment data? In other words, did the observer only collect data from horses which they knew had been treated? If so, there is a significant risk that the results are biased. This information needs to be provided in the description of the methods and the risk of bias should be discussed as a significant limitation to this study.
It appears that the study was designed with the expectation that there would be an observable difference between the response of horses with KSS and those not affected. Presumably that is why there is no control group. Although it does not unduly influence the finding that HILT brings about changes in the response to back palpation, the manuscript would be significantly improved if a control group were included. This is not as important as the problems associated with a lack of blinding in data collection.
Results.
The results are presented appropriately.
Discussion.
There are several places in the discussion where the conclusions reported could be questioned.
“The photothermal effect of HILT induces mild local hyperthermia, sufficient enough to accelerate cellular activity and vasodilatation, thus leading to enhanced local blood circulation [45]. This results in a hastened removal of inflammatory cytokines, improvement in the mitochondrial oxidation process, production of adenosine triphosphate and more efficient absorption of tissue swelling [46]. In the current study, we did not control the presence and degree of vasodilatation, although our previous research confirmed vessel diameter increase in irradiated tissue immediately after HILT [47,48]. Mild inflammation is reduced when tissue temperature increases by more than 1°C…”
Do the authors propose that the beneficial effects on pain were due to the changes described in this paragraph? If the horses were examined immediately after application of the HILT is it reasonable to expect that changes such as increased removal of cytokines or more efficient absorption of tissue swelling could have occurred?
“It can be assumed that in horses without KSS the myorelaxation effect can be longer than in horses with KSS, which (as a chronic disease) causes constant tissue irritation.”
How is this assumption justified? The underlying cause of pain in horses without KSS was not determined so they too could have had a chronic condition. Even in the horses identified as having KSS there is no certainty that this was the cause of the pain they were experiencing, or that other, concurrent conditions existed.
Reporting the findings of the study without trying to expand into areas that are not specifically relevant to the study is entirely appropriate. Provided the question of blinding can be adequately addressed the finding of reduced back pain is important and is a standalone finding.
Reviewer 3 Report
1. I advise against combining descriptions of low-intensity laser therapy treatment papers with those of high-intensity laser therapy. They are wholly diverse, disproportionate, and so incomparable results.
2. It is not suitable to describe the research of smooth muscle lesions here.
3. Mention which databases were accessed in to claim that this is the first study of HILT in muscular soreness in the back of horses, as well as which keywords were utilized in this research. Examine the text.
4. Add the evaluation time, whether it was immediate (min or hours) or days afterward, to the aim. Make the method's time apparent. Of course, a temperature increase is to be expected immediately following laser therapy, but does it linger for days?
5. Did the temperature of the horse's dorsal area drop after a few days of treatment? Is there a long-term effect, or has it simply been assessed in the short term?
6. Were horses with disc herniation or radiculopathy excluded? Explain how they were appraised instead.
7. Please let me know whether all of the animals were radiographed.
8. What kind of horses were they (breed)?
9. What is the average length of KSS illness? Inform.
10. When the thermal photos are obtained, what are the ambient temperature and humidity like?
11. Was the wind current controlled?
12. Were the animals brushed beforehand?
13. Were the animals housed under cover for thermal imaging?
14. Were all of the photos shot in the morning?
15. What is the inaccuracy (precision) of the device's temperature measurement? 1-2º C?
16. Figure 1 shows the second picture that expanded the temperature on the horse as a whole. Was there any systemic impact of the therapy that resulted in vasodilation of additional perforating cutaneous arteries (body relaxation), or was this an artifact? To explain.
17. In figure 1, the first image was not evident to the reader because the ROI was selected on the longissimus dorsi muscle, and is no longer cranial or caudal. Was that the animal's most sensitive area? Explain.
18. It was unclear why the authors used Tavg Min - Max. Strange because it is an average of T min and max of 10 horses.
19. Why didn't the authors measure the ROI's Tmin with Tmax (delta T)? The smaller the difference, the more homogenous the temperature, and hence the lower the inflammation.
20. Line 260 mentions that the laser therapy utilized produces vasodilation, but is it established that this is from the muscle or merely from the skin?
21. The heat produced by laser treatment on the skin produces muscular vasodilation, is it? Many heat treatments generate a neurological reaction and somatic cutaneous impact of relaxation.
22. What temperature might cause heat burn on the horse's skin? To cite.
23. Could thermography be used to reduce the risk of burns? Did the writers consider this utility?
24. Was there any bone heating with the laser therapy?
25. Better distinguish the photothermal effect from the photomechanical effect in the discussion language.
26. It was not obvious how heat laser treatment may cause mechanical pressure. Explain this better from a physics standpoint. The content appears to be perplexing the reader since the effect of shock waves is radically different. Wouldn't they be heat waves as opposed to pressure waves? To be corrected.
27. HILT laser treatment was not seen or sufficiently evident in this paper.
Round 2
Reviewer 1 Report
I am happy with the changes made by the authors, which address most of my concerns. There are a number of typographical errors, some introduced by the editing process, that require attention, but these are minor issues.
I would recommend a small change to the title, Introducing the phrase "A Preliminary Study Of" at the start.
Author Response
Thank you for the second review.
The phrase "A Preliminary Study Of" has been intrducing in the title.
Kind regards,
Paulina Zielinska.
Reviewer 2 Report
The authors have addressed my concerns.
Author Response
Thank you for the second review.
Kind regards,
Paulina Zielińska.